# The Risk Factors for New-Onset Calf Muscle Venous Thrombosis after Hip Fracture Surgery

**DOI:** 10.3390/jpm13020257

**Published:** 2023-01-30

**Authors:** Qianzheng Zhuang, Qifei He, Aobulikasimu Aikebaier, Wenshi Chen, Jianquan Liu, Daping Wang

**Affiliations:** 1Hand and Foot Surgery Department, The First Hospital Affiliated to Shenzhen University, Shenzhen Second People’s Hospital, Shenzhen 518000, China; 2Department of Bone Joint and Musculoskeletal Tumor, The First Affiliated Hospital of Shenzhen University, Shenzhen Second People’s Hospital, Shenzhen 518000, China; 3Shantou University Medical College, Shantou 515041, China; 4Department of Rehabilitation, The First Affiliated Hospital of Shenzhen University, Shenzhen Second People’s Hospital, Shenzhen 518060, China; 5Department of Biomedical Engineering, Southern University of Science and Technology, Shenzhen 518055, China

**Keywords:** hip fracture, calf muscle venous thrombosis, deep venous thrombosis

## Abstract

Background: Calf muscle venous thrombosis (CMVT) is among the most important medical complications after hip surgery. CMVT has been known for many years, but many opinions about the incidence and risk factors of CMVT are still controversial. The objective of this retrospective study was to investigate the incidence and associated risk factors of postoperative CMVT in patients with hip fractures. Methods: Patients with hip fractures from January 2020 to April 2022 (*n* = 320) at Shenzhen Second People’s Hospital were recruited in this study. The personal characteristics and clinical data of CMVT and no-CMVT patients were compared and analyzed. Binary logistic regression analyses were performed to identify potential risk factors of CMVT in patients with hip fractures. Finally we performed a receiver operating characteristic (ROC) curve analysis to compare the diagnostic values of different variables. Results: The overall incidence of new-onset CMVT in patients with hip fractures was 18.75% (60 of 320). Among the 60 CMVT patients, 70% (42 of 60) were diagnosed with femoral neck fractures, 28.3% (17 of 60) with intertrochanteric fractures, and 1.7% (1 of 60) with subtrochanteric fractures. No pulmonary embolism (PE) occurred. High preoperative D-dimer (OR = 1.002, 95%CI 0.97–1.03), sex (OR = 1.22, 95%CI 0.51–2.96), the caprini score (OR = 2.32, 95%CI 1.05–5.16) and the waterlow score (OR = 1.077, 95%CI 0.35–3.36) significantly increased the risk of developing postoperative new-onset CMVT. Conclusions: CMVT has become a common clinical disease, and its harm should not be underestimated. Our study found that D-dimer, sex, the caprini score and the waterlow score were independent risk factors for postoperative CMVT. According to our clinical work, we should pay attention to identifying the risk factors of CMVT formation and targeted intervention measures to prevent new-onset CMVT.

## 1. Introduction

China has the largest population of elderly people in the world, and faces the enormous challenge of increasing hip fractures every year [1]. The number of hip fractures is estimated to be 2.6 million in 2025, and 4.5 million by 2050. Management of these fractures and prevention of complications will remain an important task for healthcare systems globally [2]. Hip fracture is a fracture of the proximal femur, the upper part of the thigh, near the hip joint, and there are several types, including femoral neck fractures, intertrochanteric fractures, and subtrochanteric fractures. Hip fractures are considered to be one of the most disabling injuries in patients over the age of 60. Hip fracture is associated with high mortality, given that it is the leading cause of hospitalization and requires surgical treatment in an orthopedic trauma center [3]. There are a few studies evaluating the incidence of complications after hip fracture surgeries [4,5]. Although most patients had no postoperative complications, the inpatients with postoperative complications (PE, deep vein thrombosis (DVT), hypostatic pneumonia, etc.) had significantly higher mortality, especially those with multiple complications [6].

Patients undergoing major orthopedic surgeries are at a particularly high risk, with an estimate of 40–60% incidence of DVT without prophylaxis [7]. DVT is a common postoperative complication of hip fractures, which leads to an increase in mortality, incidence rate and nursing costs [8]. Typical clinical manifestations of DVT include swelling, pain, warmth, and redness of the affected limb. Alternatively, DVT can occur asymptomatically. Trauma, infection, peripheral arterial disease, and other venous diseases may have similar clinical features as DVT. Furthermore, DVT can coexist with any of these processes. Due to the lack of specificity of CMVT in clinical practice, once ignored, it may spread to the main deep veins or even generate PE. DVT can be divided into two categories: (1) the proximal venous thrombosis which shows venous thrombus in popliteal veins or above, including femoral, external iliac, internal iliac and common iliac veins, and inferior vena cava; (2) the distal venous thrombosis with thrombus mainly in infrapopliteal veins (including anterior and posterior tibial veins and peroneal vein) and calf muscular veins (including gastrocnemius and soleus veins) [9,10]. For the proximal DVT, it is now clear that there are two treatment reasons: prevention of clinical pulmonary embolism (PE) and prevention of post thrombotic syndrome (PTS) [11]. In contrast, the distal DVT treatment remains controversial [12]. It was widely believed that the thrombus in the calf muscle vein could be well fixed and was too small to produce clinical PE. This is the basis of the view that the distal DVT will not lead to clinical PE. However, new evidence has been found that calf muscle venous thrombosis (CMVT) produces proximal thrombus and then pulmonary embolism. With the wide application of vascular Doppler ultrasound, the detection of CMVT has become more accurate [13,14]. The harm of CMVT has become a problem that should not be ignored [15].

In traditional clinical work, before sonographers performed lower limb venous Doppler ultrasound examinations on patients for screening of DVT using a standardized method, doctors usually evaluated the patient’s condition and made corresponding diagnosis and treatment by observing the patient’s laboratory examination indicators (such as the changes of D-dimer examination results). At present, there are still no universal risk assessment models (RAMs) in the international field to evaluate the occurrence of DVT in patients after fracture surgery, so as to identify high-risk and low-risk patients and reduce the occurrence of thromboembolism. Currently the commonly used RAMs in clinical practice for evaluating venous thrombus in the lower limbs are the Well score, the Autar score, the Caprini score, and so on. Nurses in our hospital generally use the caprini score to evaluate patients on admission, postoperatively, and at discharge. In our study, we aimed to investigate the risk factors for new-onset CMVT after hip fracture surgery, so we included the caprini score collected by nurses at admission. In our study we not only focus on laboratory indicators, such as D-dimer (previous studies have shown that D-dimer is an independent risk factor for DVT formation), but also pay more attention to other indicators similar to the caprini score. We have tried our best to include those factors that haven’t been paid attention to in previous studies, such as the Waterlow score (used in many hospitals to stratify the risk of decubitus ulcer development), the American Society of Anesthesiologists (ASA) grade (which forms part of the World Health Organization preoperative checklist, mandated in the UK National Health Service (NHS) hospitals), the New York Heart Association (NYHA) functional classification (a fundamental tool for risk stratification of heart failure (HF) and to develop different treatment regimens based on their grading), the Mallampati score (introduced in the 1980s as a way to predict the difficult airway).

Though there are many reports on DVT after hip fracture surgeries, there are very few studies on systemic analysis of CMVT [16,17]. In this study, we focused on the new-onset CMVT after hip fracture surgeries through data collation. The purpose of this study was to evaluate the influencing factors of CMVT after hip fractures, and to provide supporting evidence for clinical prevention and treatment of CMVT.

## 2. Methods

### 2.1. Patient Selection

We conducted a retrospective study on patients who underwent hip surgeries at Shenzhen Second People’s Hospital between January 2020 and April 2022. Data were collected and analyzed according to the exclusion criteria (Figure 1). The exclusion criteria for patients were: conservative treatment, preoperative DVT (including CMVT), pathologic fractures, old fractures (>21 d from injury), incomplete data. The ethics committee in the hospital approved the study and all participants included in this study provided informed consent.

### 2.2. Definition of CMVT

We did ultrasound examinations of lower limbs immediately after admission. Before ultrasound examination, we did not use drugs, such as rivaroxaban and low molecular weight heparin, to prevent CMVT, because we performed emergency surgeries to avoid the risk of surgical bleeding. If we find a DVT patient, we will use low molecular weight heparin for anticoagulation.

Trained and certified sonographers performed lower limb venous Doppler ultrasound examinations on patients before and 1 to 2 days after surgery for screening of CMVT using a standardized method. We recorded the CMVT scores in gastrocnemius and soleus veins.

### 2.3. Data Collection

Inpatient medical records were retrieved to collect relevant clinical and laboratory data. The detailed information was related to demographics (age, sex), BMI (calculated by body weight divided by height square), comorbidities (hypertension, diabetes, chronic heart disease, cerebrovascular disease, lung disease, liver disease, renal insufficiency, peripheral vascular disease, etc.), injury related data (fracture type), previous surgical history, preoperative D-dimer, the caprini score, the NYHA classification, the Mallampati score, the ASA score, the waterlow score, anesthesia method, operative procedures, intraoperative infusion volume, postoperative blood pressure (SBP, DBP).

### 2.4. Statistical Analysis

The observed data are statistically analyzed using SPSS26.0 statistical software (IBM, New York, NY, USA). We divided the patients into two groups: patients diagnosed with DVT as the cases group and those without DVT served as the control group. The measurement data are all expressed as mean ± standard deviation or percentage. Student *t*-test or Mann–Whitney test was used for all continuous variables(age, BMI, preoperative D-dimer, the caprini score, the waterlow score, intraoperative infusion volume, postoperative blood pressure (SBP, DBP)). All categorical variables (sex, comorbidities (hypertension, diabetes, chronic heart disease, cerebrovascular disease, lung disease, liver disease, renal insufficiency, peripheral vascular disease, etc.), injury related data (fracture type), previous surgical history, the NYHA classification, the Mallampati score, the ASA score, anesthesia method, operative procedures) were analyzed by chi-square or Fisher’s exact test. After correlation analysis of all data, binary logistic regression model was used to distinguish the independent predictors of DVT. We carried out a receiver operating characteristic (ROC) curve analysis to compare the diagnostic values of different variables. Statistical significance was defined as *p* < 0.05.

## 3. Results

Demographic and clinical characteristics of patients by CMVT status are presented in Table 1. Over half (67%) of the 320 patients included in the study were male. The patients diagnosed with CMVT were 45 males (75% of all confirmed CMVT cases) and 15 females (25% of all confirmed CMVT cases). The average age of patients diagnosed as new-onset CMVT after hip fracture surgery is 75.38 years. The average age of patients without CMVT after hip fracture surgery is 75 years. The average infusion volume during operation is 1370 mL. The mean duration of surgery for patients who received operations for new-onset CMVT was 86.12 min. Among the 60 patients with diagnosed CMVT after surgery, 26 (43.3%) had no complications, 22 (36.7%) had 1–2 complications, and 12 (20%) had three or more complications. Among the 60 CMVT patients, the ASA scores I–II were present in 40% patients (24), and the rest 60% of patients (36) had the ASA scores of III–IV. Among the 60 CMVT patients, 90% (54) of patients were in the NYHA classification I–II, and the rest 10% (6) of patients were in the classification III–IV. Patients with new-onset CMVT had a mean SBP of 124.4 mmHg and a mean BDP of 66.37 mmHg. There are four operative procedures: total hip replacement (THA), hemiarthroplasty, proximal femoral nail antirotation (PFNA), open reduction and internal fixation (ORIF). Among the 60 patients diagnosed with CMVT, 43.3% (26) were operated on THA, 30% (18) were operated on hemiarthroplasty, 25% (15) were operated on PFNA, and 1.7% (1) were operated on ORIF. The overall incidence of new-onset CMVT in patients with hip fractures was 18.75% (60 of 320). Among the 60 CMVT patients, 70% (42 of 60) were diagnosed with femoral neck fractures, 28.3% (17 of 60) with intertrochanteric fractures, and 1.7% (1 of 60) with subtrochanteric fractures. No pulmonary embolism (PE) occurred. There was no significant difference in terms of prevalence of age, time to surgery, infusion volume, comorbidity, previous surgical history, types of anesthesia, operative procedures, the waterlow score, the NYHA score, the ASA score, the Mallamptis score, hypertension, diabetes, cardiovascular disease, BMI, SBP, DBP, between patients with and without CMVT.

The results of the binary logistic regression analysis were shown in Figure 2 for postoperative new-onset CMVT. We found that higher preoperative D-dimer (OR = 1.002, 95% CI 0.97–1.03), sex (OR = 1.22, 95% CI 0.51–2.96), the caprini score (OR = 2.32, 95% CI 1.05–5.16) and the waterlow score (OR = 1.077, 95% CI 0.35–3.36) significantly increased the risk of developing postoperative new-onset CMVT. The receiver operating characteristic curve (ROC) is commonly used to assess the diagnostic value of different methods: the larger the area under the curve (AUC), the better the diagnostic value. In the study, we calculated the AUC of independent risk factors and found that the caprini score had the highest diagnostic value (AUC, 0.593; 95%CI, 0.520 to 0.667; *p* = 0.019). However, for other variables, no significant difference could be identified and the AUC analysis results were not significant (*p* > 0.05) (Figure 3 and Table 2).

## 4. Discussion

Elderly hip fracture is dubbed as the ‘life’s last fracture’. In clinical work, we found that older adults without anticoagulation had a very high rate of new-onset CMVT after a hip fracture surgery. CMVT is a disorder of venous return caused by abnormal blood coagulation in the venous of the lower extremities, which completely or incompletely obstruct the blood vessels. CMVT is a common complication after hip fracture surgery. Therefore, CMVT has attracted extensive attention from clinical medical workers. In this study, the incidence of new-onset CMVT in patients with hip fractures was 18.8%, and no patient was diagnosed with pulmonary embolism. The results of the binary logistic regression analysis showed that sex, D-dimer, the caprini score, and the waterlow score were independent risk factors for the occurrence of CMVT (Figure 2). Then we analyzed each variable with ROC curve, and found that only the Caprini score was statistically significant. (AUC, 593; 95% CI, 0.520 to 0.667; *p* = 0.019) (Figure 3 and Table 2). Early prevention and intervention of these risk factors for CMVT are clinically required to reduce the occurrence of CMVT.

According to the classical thrombosis theory, the three factors of thrombosis include slow blood flow, hypercoagulable state and vascular injury [18]. The risk of venous thrombosis in men is higher than that in women, which has triggered a debate on whether there is a sex difference in the risk of venous thrombosis [19]. A genome-wide gene-centric study showed that men had a significantly higher risk of the first-time venous thrombosis than women [20]. The risk of first and recurrent venous thrombosis is higher in men than in women. However, the pathophysiology behind this phenomenon is not clear [21]. In our study, we found that sex (OR = 1.22, 95% CI 0.51–2.96) was an independent risk factor for the occurrence of CMVT. However, the mechanism of sex and CMVT formation needs further investigation.

At present, ultrasound and preoperative D-dimer are the major non-invasive diagnostic methods to exclude suspected symptomatic DVT include [22]. D-dimer is a fibrin degradation product, a fragment detectable in blood after a small protein blood clot is degraded by fibrinolysis. Plasma D-dimer increases in patients with venous thrombosis. It is a sensitive marker of thrombosis, but it is lack of specificity. D-dimer testing is only used to exclude DVT, and a positive result is not diagnostic because many conditions, such as persistent blood loss, impaired renal function, pregnancy, and atrial fibrillation, can boost D-dimer levels [23,24]. There is good clinical evidence that a negative D-dimer test result in combination with a low pretest probability determined by well validated clinical prediction rules is sufficient to exclude DVT. In such cases, further testing is unnecessary even in patients with a previous DVT [25]. Many studies on DVT in patients with lower limb fractures have now shown that preoperative D-dimer is an independent risk factor for DVT [26,27,28]. We confirm this point in our analysis in this study. We calculated AUCs for all independent risk factors, however, the AUC for D-dimer was only 0.585 (95% CI, 0.460 to 0.710; *p* = 0.17) (Figure 3 and Table 2). Our study is retrospective and limited by sample size, which may require more prospective experiments to further validate this finding. The preoperative D-dimer test is a simple, convenient and meaningful test which should be given a high priority to patients with elevated preoperative D-dimer.

We should not only focus on laboratory or imaging examinations, but also some nursing scores. In recent years, scholars have made efforts to establish risk assessment models to identify high-risk and low-risk patients in order to reduce the incidence of thromboembolic events. Caprini RAM is perhaps the most widely used and well validated risk prediction tool to date. It has been implemented in many institutions and integrated into electronic medical records [29]. It has now been shown that the 2010 Caprini RAM can effectively stratify hospitalized Chinese patients into DVT risk categories based on individual risk factors [30]. Caprini RAM has been frequently used in many other surgical fields, including general, vascular, plastic, urologic, and head and neck surgeries and may be applicable to orthopaedic surgeries as well. Is the Caprini score predictive of DVT events in orthopaedic fracture patients? The answer is yes. Many studies have shown that caprini RAM may become an important tool for orthopaedics doctors to guide DVT risk stratification and management [31,32,33]. The caprini RAM is currently a clinically valid and simple, feasible, and cost-effective DVT risk prediction tool. The caprini thrombus risk assessment model assigns a score based on 40 risk factors for DVT, such as age, BMI, medical history, surgical history, laboratory indexes, and time spent in bed, which enable precise screening of high—and very high-risk cases. Each risk factor confers a score of 1 to 5 based on the degree of risk, and each patient is categorized accordingly into four grades, i.e., low (0 to 1 point), intermediate (2 points), high risk (3 to 4 points) and very high risk (≥5 points). Compared with conventional intervention strategies, intervention strategies based on the caprini thrombus risk assessment model are more targeted, planned, and predictive, resulting in a significant reduction in the incidence of postoperative DVT through real-time tracking of DVT information, dynamic assessment of risk levels, and individualized formulation and adjustment of intervention protocols. In our study, the caprini score was an independent risk factor for postoperative new-onset CMVT. We analyzed the caprini score by ROC curve, and the test result was statistically significant by AUC analysis (AUC, 593; 95% CI, 0.520 to 0.667; *p* = 0.019) (Figure 3 and Table 2). Therefore, caprine RAM is highly sensitive and specific to patients undergoing a hip fracture surgery, which may help us manage different patients with personalized strategies.

Preoperative risk prediction before major surgery can help formulate targeted intraoperative and postoperative care. The Waterlow score (Ws) as a pre-operative risk prediction tool has been established and validated in the context of general surgery. Ws was first developed in the mid-1980s. It is used widely to stratify the risk of decubitus ulcer development among the inpatient population [34]. This is a composite score that reflects the patient’s general condition and comorbidities. It consists of 11 factors: sex, age, skin types, BMI, tissue malnutrition, exercise ability, appetite, defecation control ability, nervous system diseases, surgery/trauma and drug treatment [35]. Using a multisystem approach, based on multiple variable weighted scores, WS classifies patients into non-risk (with the score of less than 10), low-risk (with the score of 10–14), high-risk (with the score of 15–19), or very high-risk (with the score of 20 or above). Ws is mostly collected regularly by nurses in many hospitals. Previous studies have found that Ws may be an effective tool for predicting postoperative morbidity and mortality [36,37]. It can also stratify perioperative risks to compare surgical outcome data. No previous article has studied the relationship between Ws and postoperative new-onset CMVT. We found that the Ws can predict the new-onset CMVT after surgery, which is an interesting result. There are some limitations to consider. To date, the limited evidence on WS in surgical patients is mainly derived from retrospective studies. Furthermore, although WS has the advantage of being almost universally used in nursing practice, it may not be applicable to immediate or emergency situations because of the lag between admission and pressure ulcer assessment. In our study, we analyzed the Waterlow score with ROC curve, and the test results were not statistically significant with AUC analysis (AUC, 0.474; 95% CI, 0.398 to 0.550; *p* = 0.515) (Figure 3 and Table 2). Therefore, large-scale prospective studies are needed to validate these findings before they can be used as surgical risk predictors.

The major advantage of our study is that we have systematically analyzed the risk factors of new-onset CMVT after hip fracture surgery, which has been ignored in previous studies. This study excluded patients with preoperative DVT. In addition, the data of this study is based on the orthopaedic database of Shenzhen Second People’s hospital. All these will help to improve the reliability and accuracy of the results of this study. However, our study does have some limitations. Firstly, all the data we collected were extracted from one hospital, which limits its effectiveness in other populations. Further multicenter randomized controlled trials are needed. Secondly, we did not collect all the laboratory indicators of patients before and after operations. There might be more meaningful indicators besides D-dimer. If we include more variables and larger sample sizes, we can make the Nomogram to predict the new-onset CMVT after hip fracture surgery. In the nomogram, each value level of each influential factor was given a score, and individual scores were then summed to yield a total score, which was finally used to calculate a predictive value for that individual outcome event through a functional transformation relationship between the total score and the probability of occurrence. Finally, one limitation of this study is that we only studied the incidence of new-onset CMVT in inpatients during hospitalization. Since some new-onset CMVT occured after discharge, our study may underestimate the true incidence of new-onset CMVT after surgery.

## 5. Conclusions

CMVT is a common complication of hip fractures in the elderly. Our study found that D-dimer, sex, the caprini score and the waterlow score were independent risk factors for postoperative CMVT. In clinical work, we sometimes do not focus on the Caprini score and Waterlow score. With our study, in future clinical work we should pay more attention to the Caprini score and Waterlow score. These scores are important for the diagnosis and treatment of CMVT. We should not ignore the harm of CMVT. Although the presence of CMVT has been known for many years, the treatment of CMVT is still controversial. We need to make corresponding treatment plans according to the specific conditions of each individual patient.

## Figures and Tables

**Figure 1 jpm-13-00257-f001:**
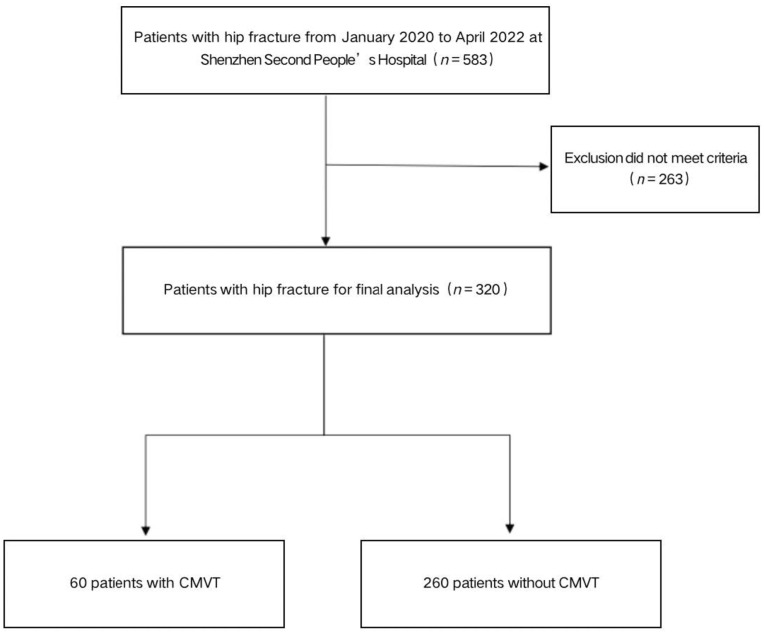
The flow chart for the selection of study participants.

**Figure 2 jpm-13-00257-f002:**
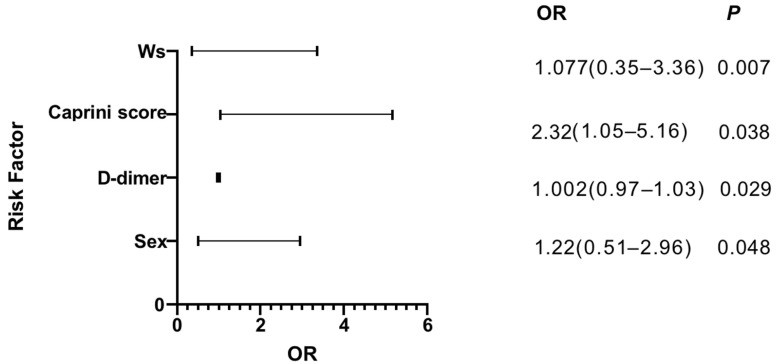
The results of the binary logistic regression analysis.

**Figure 3 jpm-13-00257-f003:**
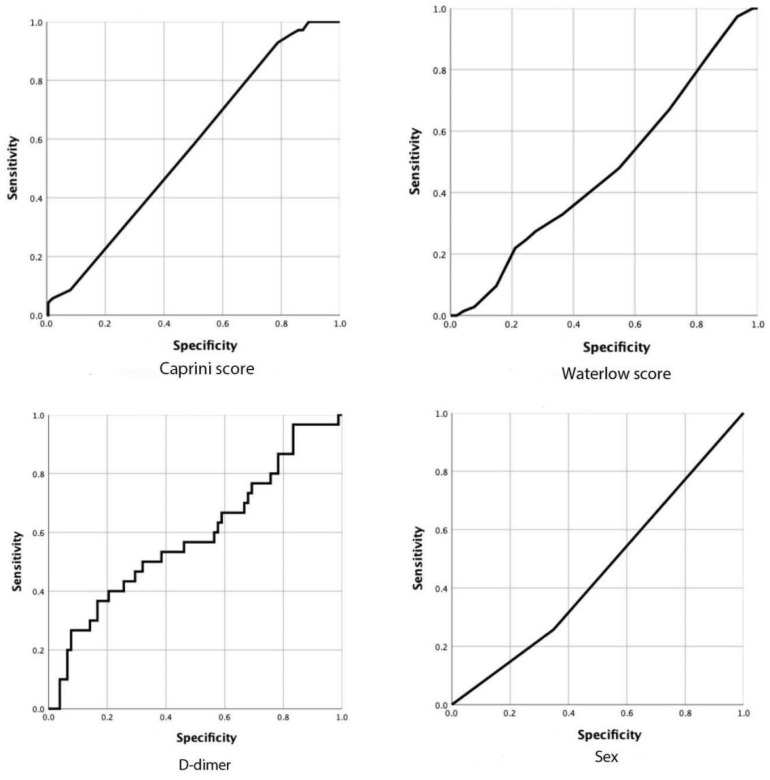
The ROC curves of different factors.

**Table 1 jpm-13-00257-t001:** Demographic and clinical characteristics of patients.

Variable		Patients with CMVT (*n* = 60)	Patients without CMVT (*n* = 260)	*p* Value
Age		75.38 (12.32)	75.00 (12.32)	0.254
Time to surgery		1.63 (1.60)	1.65 (1.531)	0.937
Duration of surgery		86.12 (40.38)	88.9 (36.72)	0.604
Waterlow score		16.67 (3.17)	16.97 (3.45)	0.53
Infusion volume		1370.0 (432.28)	1269.23 (438.84)	0.108
SBP		124.42 (19.32)	122.01 (17.71)	0.352
DBP		66.37 (9.61)	65.22 (8.57)	0.353
Caprini score		8.65 (1.40)	7.89 (2.62)	0.03
Preoperative D-dimer		5.44 (9.04)	5.16 (11.19)	0.881
BMI		22.03 (3.39)	22.13 (3.99)	0.867
Sex				0.153
	Male	45 (75.0%)	170 (65.4%)	
	Female	15 (25.0%)	90 (34.6%)	
Diagnosis				0.8
	Femoral neck fractures	42 (70.0%)	181 (69.6%)	
	Intertrochanteric fractures	17 (28.3%)	77 (29.6%)	
	Subtrochanteric fractures	1 (1.7%)	2 (0.8%)	
Hypertension		25 (42.3%)	107 (41.2%)	0.942
Diabetes		12 (20.0%)	48 (18.5%)	0.783
Cardiovascular		8 (13.3%)	26 (10.0%)	0.45
Comorbidity				0.395
	0	26 (43.3%)	89 (34.2%)	
	1–2	22 (36.7%)	128 (49.2%)	
	≥3	12 (20.0%)	33 (16.5%)	
Previous surgical history		23 (38.3%)	109 (41.9%)	0.611
Type of Anesthesia				0.614
	General anesthesia	9 (15.0%)	29 (11.2%)	
	Epidural anesthesia	43 (41.3%)	231 (88.8%)	
Operative procedures				0.845
	THA	26 (43.3%)	112 (43.1%)	
	Hemiarthroplasty	18 (30%)	67 (25.8%)	
	PFNA	15 (25.0%)	78 (30.0%)	
	ORIF	1 (1,7%)	3 (1.2%)	
NYHA				0.177
	I–II	54 (90.0%)	250 (92.3%)	
	III–IV	6 (10.0%)	20 (7.7%)	
ASA				0.824
	I–II	24 (40.0%)	113 (43.5%)	
	III–IV	36 (60.0%)	147 (56.6%)	
Mallamptis				0.742
	I–II	41 (68.3%)	192 (73.8%)	
	III–IV	19 (31.6%)	68 (25.1%)	

**Table 2 jpm-13-00257-t002:** The ROC results of different factors.

Variables	AUC	95%CI Lower Bound	95%CI Upper Bound	*p* Value
Caprini score	0.593	0.520	0.667	0.019
Waterlow score	0.474	0.398	0.550	0.515
Sex	0.503	0.425	0.581	0.943
D-dimer	0.585	0.460	0.710	0.17

## Data Availability

The datasets used and analyzed during the current study available from the corresponding author on reasonable request. We simply extracted data and did not involve the private information of patients.

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
