# Peer review of "The Risk Factors for New-Onset Calf Muscle Venous Thrombosis after Hip Fracture Surgery"

_jpm, 2023, doi:10.3390/jpm13020257_

Round 1
Reviewer 1 Report
The study presents a retrospective study on patients with hip fractures undergoing surgery at a specific hospital between January 2020 and April 2022. The study found that the overall incidence of CMVT in these patients was higher and identified four potential risk factors for CMVT: high preoperative D-Dimer, sex, Caprini score, and Waterlow score.
I have some comments, as highlighted below:
Abstract
The word "perfomed" should be "performed" as it is spelled incorrectly.
The word "cmvt" should be written as "CMVT" as it is an acronym.
The word "its" in the last sentence should be "it's" as it is a contraction of "it is".
Introduction
The word "this" in the phrase "This fracture is associated with high mortality" is unnecessary and should be removed or rephrased.
The phrase "inpatients with postoperative complications had significantly higher mortality" is awkwardly phrased and should be rewritten for clarity.
The abbreviation "DVT" should be written as "deep vein thrombosis" in the first instance it appears in the text. Even if this is addressed in the abstract
The abbreviation "PE" should be written as "pulmonary embolism" in the first instance it appears in the text. Even if this is addressed in the abstract
The abbreviation "CMVT" should be written as "calf muscle venous thrombosis" in the first instance it appears in the text. Even if this is addressed in the abstract
The phrase "proximal thrombus" should be defined or explained earlier in the text.
The phrase "vascular Doppler ultrasound" should be defined or explained earlier in the text.
The phrase "new-onset CMVT after hip fracture sugery" should be written as "new-onset CMVT after hip fracture surgery" as "sugery" is spelled incorrectly.
Methods
The phrase "body mass index" should be abbreviated as "BMI" in all instances.
The word "infusion" should be spelled as "infusion volume" as it is written incorrectly.
Results
Figure 1. Please provide a better-quality image here. The text seems like scanned from a file. Please make it look more professional
Discussion
The word "hypercoagulable" is misspelled as "hypercoaguliable".
The word "noninvasive" should be spelled as "non-invasive".
The abbreviation "D-dimer" should be written as "D-dimer" in all instances.
Conclusions
This section should include the main findings of the study and be highlighted. Current form is too general.
Author Response
We would like to thank JPM for giving us the opportunity to review our manuscript. carefully addressed, and we have submitted our revised manuscript. We thank the assigned editor for the careful, meticulous, and responsible work. We are grateful to the reviewer for the insightful comments. We also hope to receive your reply.
Best wishes.
Sincerely,
Qianzheng Zhuang

Reviewer 2 Report
An interesting report. Did patients found with CMDVT have symptoms? You may discuss.
Author Response

(The authors gave the same response as above.)
